# Spectral Analysis of Light-Adapted Electroretinograms in Neurodevelopmental Disorders: Classification with Machine Learning

**DOI:** 10.3390/bioengineering12010015

**Published:** 2024-12-28

**Authors:** Paul A. Constable, Javier O. Pinzon-Arenas, Luis Roberto Mercado Diaz, Irene O. Lee, Fernando Marmolejo-Ramos, Lynne Loh, Aleksei Zhdanov, Mikhail Kulyabin, Marek Brabec, David H. Skuse, Dorothy A. Thompson, Hugo Posada-Quintero

**Affiliations:** 1Caring Futures Institute, College of Nursing and Health Sciences, Flinders University, Adelaide 5000, SA, Australia; lynne.loh@flinders.edu.au; 2Biomedical Engineering Department, University of Connecticut, Storrs, CT 06269, USA; javier.pinzon_arenas@uconn.edu (J.O.P.-A.); luis.mercado_diaz@uconn.edu (L.R.M.D.); hugo.posada-quintero@uconn.edu (H.P.-Q.); 3Behavioural and Brain Sciences Unit, Population Policy and Practice Programme, UCL Great Ormond Street Institute of Child Health, University College London, London WC1N 1EH, UK; irene.lee@ucl.ac.uk (I.O.L.); d.skuse@ucl.ac.uk (D.H.S.); 4College of Psychology and Education, Flinders University, Adelaide 5000, SA, Australia; fernando.marmolejoramos@flinders.edu.au; 5“VisioMed.AI”, Golovinskoe Highway, 8/2A, 125212 Moscow, Russia; zhdanov@visiomed.ai; 6Pattern Recognition Lab, Friedrich-Alexander-Universität Erlangen-Nürnberg, 91058 Erlangen, Germany; mikhail.kulyabin@fau.de; 7Institute of Computer Science of the Czech Academy of Sciences, Pod Vodarenskou Vezi 2, 182 00 Prague, Czech Republic; mbrabec@cs.cas.cz; 8National Institute of Public Health, Srobarova 48, 100 00 Prague, Czech Republic; 9The Tony Kriss Visual Electrophysiology Unit, Clinical and Academic Department of Ophthalmology, Great Ormond Street Hospital for Children NHS Trust, London WC1N 3BH, UK; dorothy.thompson@ucl.ac.uk; 10UCL Great Ormond Street Institute of Child Health, University College London, London WC1N 1EH, UK

**Keywords:** biomarker, retina, autism, attention deficit hyperactivity disorder, sex, medication, feature selection

## Abstract

Electroretinograms (ERGs) show differences between typically developing populations and those with a diagnosis of autism spectrum disorder (ASD) or attention deficit/hyperactivity disorder (ADHD). In a series of ERGs collected in ASD (*n* = 77), ADHD (*n* = 43), ASD + ADHD (*n* = 21), and control (*n* = 137) groups, this analysis explores the use of machine learning and feature selection techniques to improve the classification between these clinically defined groups. Standard time domain and signal analysis features were evaluated in different machine learning models. For ASD classification, a balanced accuracy (BA) of 0.87 was achieved for male participants. For ADHD, a BA of 0.84 was achieved for female participants. When a three-group model (ASD, ADHD, and control) the BA was lower, at 0.70, and fell further to 0.53 when all groups were included (ASD, ADHD, ASD + ADHD, and control). The findings support a role for the ERG in establishing a broad two-group classification of ASD or ADHD, but the model’s performance depends upon sex and is limited when multiple classes are included in machine learning modeling.

## 1. Introduction

Exploring biomarkers in neurodevelopmental disorders has been the focus for many groups interested in identifying characteristic biological features that identify clinical populations [1]. To contribute to this field, this paper explores the signal analysis of the electroretinogram (ERG) waveform combined with machine learning (ML) tools to develop a classification model for individuals with autism spectrum disorder (ASD), attention deficit/hyperactivity disorder (ADHD) and those with co-occurring clinical diagnoses of ASD + ADHD. The retina is viewed as a ‘window to the brain’ [2], with differences observed in the shape of the ERG in neuropsychiatric disorders including schizophrenia/bipolar disorder [3], Alzheimer’s Disease [4], depression [5], and Parkinson’s Disease [6]. For a description of the physiology of the ERG, see Appendix B (following the discussion section) and recent reviews [7,8,9].

The origins of harnessing retinal signals as potential biomarkers in ASD date back to 1988 when Edward Ritvo first explored the dark-adapted ERG responses in *n* = 27 children and adults with ASD and found a reduced b-wave amplitude in approximately half the participants, which suggested a glutamate signaling pathway deficit in this group [10]. Ritvo also performed a small pilot study in a family but did not identify a strong familial link with the ERG responses [11]. It was much later that these initial findings were extended by Constable et al. (2016) [12] in a small adult population of high-functioning adults with ASD, which supported these initial findings and further identified reductions in the light-adapted ERG b-wave amplitudes and the shape of the oscillatory potentials (OPs).

The first multicenter study to be performed in (*n* = 90) ASD and (*n* = 87) control children revealed reduced a- and b-wave amplitudes at the higher flash strength; however, the study did not explore the OPs in detail [13] but found the PhNR was also normal, suggesting normal retinal ganglion cell function [14]. One observation made during this study was that some children who presented had a co-occurring diagnosis of ASD + ADHD, and it was noted that this sub-group had elevated b-wave levels compared to the children with a sole ASD diagnosis. This led to an exploratory study in (*n* = 15) ADHD participants that revealed a large b-wave amplitude in this group, which differentiated ADHD from ASD [15]. The signal analysis of the available waveforms showed higher energy levels in the ERG signal in the ADHD participants in this study [16]. These findings supported the conclusions of Lee et al. (2022) [15] that the pattern of differences in the ERG signal was most likely related to differences in the balance of GABA and glutamate signaling in ASD and ADHD. However, Friedel and colleagues failed to replicate the findings of a reduced b-wave in (*n* = 32) ASD adults using the same ERG recording protocol [17]. Huang et al. (2024) [18] also reported no significant difference in the b-wave amplitude in ASD adults, did report but a larger a-wave which was reduced following exposure to a GABA(B) agonist, suggesting the ERG may provide a possible pathway to monitor drug efficacy in ASD [18].

With respect to ADHD, there is evidence of greater background retinal ‘neural noise’ in (*n* = 20) adults. This correlates with inattention scores [19], suggesting a functional change in the retina of ADHD individuals. The most recent ERG findings in (*n* = 27) ADHD adults did not replicate the elevated b-waves previously reported but instead highlighted a reduced light-adapted a- and b-wave amplitude and delayed the b-wave time to peak amongst the female ADHD, participants which suggested some sex difference within the ADHD population [20]. Taken together, the different studies have reported contrary results at times, which may be due to several factors such as the diagnostic procedures at different locations, sex, or the age of the populations. Until more recently, most studies have relied upon the time domain features of the ERG such as the amplitude and time to peak of the principal components or the shape and peak of the photopic hill and its mathematical characterization [21]. Methylphenidate, used to control ADHD symptoms and elevate central dopamine levels, also induces changes in the ERG amplitude in some participants [22].

ERG is conventionally analyzed using the main time domain parameters that relate to the amplitude of the two main a- and b-wave peaks and the time points at which the peaks occur [23]. However, relying on these markers limits the number of features that ML models can use to improve classification between groups. Therefore, signal analysis in the time–frequency domain is utilized to decompose the signal into multiple frequency-based sub-bands [16,24,25,26], which expands the number of features available for ML models. The time–frequency analysis of the ERG waveform was developed by Gauvin and colleagues, who decomposed the ERG signal using a discrete wavelet transform (DWT) with a Haar mother wavelet [27,28,29,30]. An alternative signal analytical approach using variable-frequency complex demodulation (VFCDM) [31] was applied to the ERG waveforms and compared to DWT with the introduction of ML for group classification, with the model achieving a sensitivity of 0.85 and specificity of 0.78 [24,26].

ML using visual electrophysiological signals has been applied to pattern ERG to support therapeutic intervention decision making [5]. Applications using ML have also been used to identify the risk of hydroxy chloroquine retinopathy, associated with the analysis of multifocal ERG [32]. Murine models of glaucoma have also employed ML models of the full-field ERG to support the early diagnosis of glaucoma [33]. Several ML models have also been used with ERG and genotypes to identify individuals that are most likely to benefit from gene therapies in *ABCA4* retinopathies [34] and visual function [35]. Neurodevelopmental disorders, such as ASD and ADHD, share an overlap in traits, such as in responding to novel environments [36], with some common genetic and neuroanatomical differences reported [37]. There is a frequent co-diagnosis of ASD and ADHD within individuals, which highlights the need to identify potential markers that may be specific to either ASD or ADHD or the ASD + ADHD phenotype [38].

Several studies have explored ML models to improve biomarker discovery in neurodevelopmental disorders, a comprehensive review of which is beyond the scope of this study. A review of imaging studies reports accuracies between 48.3 and 97% for ASD classification, as determined by heterogeneity in the study population [39,40]. Attempts have been made to address this issue using the Autism Brain Imaging Data Exchange dataset acquired from several sites with an area under the curve of 79% obtained based on brain network analysis [41]. Eye movement studies based on gaze patterns in ASD with ensemble ML algorithms report an F1 score of 95.5% and provide a fast and user-friendly method for ASD classification [42]. Using Shapley feature selection, eye tracking has been successfully used in children under three years of age with the Random Forest classifier to obtain an area under the curve of 0.85 [43]. Electroencephalogram recordings also yielded classification accuracies of 98% when neural networks were employed to classify ASD [44]. The limitations of these studies include the way that the models support the classification of two groups, such as ASD vs. control, and do not consider other neurodevelopmental disorders or individuals that meet an ASD and ADHD diagnosis and so the classification accuracies are limited to one class.

Contributing to this field, this descriptive analysis aims to illustrate the potential of ML applications with ERGs to improve classification in a range of neurodevelopmental disorders. We assessed the influence of sex assigned at birth due to the reported differences in female ADHD individuals [20]. This study includes diagnostic groups, including ASD, ADHD, and individuals that meet both ASD and ADHD diagnostic criteria, to elucidate if ML models with feature refinements in feature selection can classify across the three categories from a typically developing population. As such, it provides insights into the strengths and weaknesses of ML models for this task.

## 2. Materials and Methods

### 2.1. Electrophysiology

Visual electrophysiology was performed in accordance with the International Society of Clinical Electrophysiology of Vision guidelines [23]. In all cases, a white flash custom Troland test protocol was used to compensate for pupil diameter, with the right eye always recorded first. The RETeval (LKC Technologies Inc, Gaithersburg, MD, USA) handheld ERG unit was used in all cases with adult skin electrodes. The reported raw waveforms, the digital infra-red images of the eye being tested, and time domain parameters were exported using RFF Extractor software (LKC Technologies Inc., Gaithersburg, MD, USA). Traces were rejected from the reported waveform average if they fell above or below the 25th percentile of the average due to blinks or the loss of fixation artifacts. The sampling frequency was 2 kHz, with bandpass filters of 0.1–300 Hz. Flash strength was reported in Troland seconds (Td.s).

In study 1, the test protocol consisted of 9 randomized flash strengths recorded on a white 40 cd.m^−2^ background with 60 averages to generate the reported ERG waveform. In study 2, a shortened test protocol was used with two flash strengths (113 Td.s and 446 Td.s) on a 40 cd.m^−2^ white background with 30 averages to generate the reported average ERG waveform. The aim of this protocol was to reduce the test time by reducing the number of averages and to use the two strengths that were most likely to differentiate ADHD from controls based on our early reported findings [13,14,15]. In this case, the 113 Td.s result was always recorded, first followed by the 446 Td.s result, in the right and then left eye. See Appendix A for further details about flash strengths and recording protocols.

### 2.2. Electroretinogram Analysis

Three analyses of the ERG waveform ‘signal’, in both the time and time–frequency domains, were conducted with the aim of identifying the best set of features to differentiate the control, ASD, ADHD, and the ASD + ADHD groups. Further details of the signal analysis methods are provided in Appendix C (following the discussion section) and were previously described in detail using discrete wavelet transform (DWT) [16,27,28,29,30] and variable-frequency complex demodulation (VFCDM) [24,25,31,45].

### 2.3. Machine Learning

A systematic evaluation of seven ML techniques was conducted. This included Random Forest (RF), Adaptive Boosting (AdaB), Gradient Boosting (GradB), Extreme Gradient Boosting (XGB), Support Vector Machine (SVM), K-nearest neighbor (KNN), and multi-layer perceptron (MLP). This was conducted to consider the great variety of nonlinear and complex patterns that each technique can learn and the variation in performance that this may imply. The training of each model was divided into three stages (dataset selection, hyperparameter optimization, and model training), with each stage outlined in Figure 1.

**Stage 1:** *Dataset selection* involved the implementation of feature fusion to decide on which combination of the available features to use from the three analysis methods. One combination was the fusion of the time domain (TD) features with either DWT or VFCDM features. A second involved was the concatenation of all the features (TD + DWT + VFCDM). A third one was a combination of selected features, which included all of them except the a-wave time, the OP160 coefficients, and the statistical features of the 7th and 8th VFCDM components [26]. The second step involved the selection of the site from which the samples were obtained (site 1 = Flinders; site 2 = UCL; site 3 = both). The third step consisted of concatenating the features based on the eyes (right or left) and the flash strength used. These selections were made using a single eye and flash strength, by combining the features acquired from the two eyes using the same flash strength, and by combining two flash strengths using either the same eye or different eyes. It should be noted that the concatenated instances were from the same subject (participant) and no repetition of the same signal was employed in the combinations. For example, if one participant had replicate measurements of the ERG in an eye, then only one sample was used from that eye.

**Stage 2:** *Hyperparameter optimization* is the process of identifying the optimal hyperparameter values for the ML models. This is typically accomplished by training and testing the model on multiple subsets, varying the values, and selecting the ones that yield the best performance. For this work, the hyperparameter optimization of the ML model from the complete selected dataset was conducted using a 3-fold subject-wise cross-validation approach with synthetic minority oversampling (SMOTE) [46] and RF-based feature selection, with the threshold value exceeding 0.25 × the mean of all the *feature importance* (*FI*) values. In this context, a feature is defined as a measurement obtained from the ERG signal through TD or time–frequency domain analysis. The SMOTE balancing technique was implemented to mitigate potential bias introduced by the dataset’s imbalance. The *FI* represents a relevance estimation for each of the features used for the model’s prediction, where a high score means that the feature has a bigger impact on the prediction, while small scores are given to the features that do not affect the prediction greatly. Feature selection was utilized to eliminate non-informative features while preserving the most important features based on the *FI* scores. Features from one recording of each subject were included in either the training or the test dataset but never both.

**Stage 3:** *Model training* and validation used a 10-fold subject-wise cross validation in conjunction with the hyperparameters and features selected from stage 2. In clinical prediction models, cross validation should be performed on a subject basis for a meaningful evaluation, as in our study, where multiple samples existed per subject. This helps to improve the generalization and result interpretation of the model [47]. In addition, this prevents data leakage, when a subject’s data are used for both training and testing. Furthermore, due to the limited number of instances for certain groups (such as ASD + ADHD) and to ensure sufficient variation within the training set, k = 10 was selected for the k-fold cross validation. At each fold, the training set was subjected to SMOTE balancing to maintain the same number of instances per group during the training. However, the test set kept the unbalanced distribution. As a result of the imbalance problem, the F1 score and balanced accuracy (BA) were selected as the most appropriate evaluation metrics with which to evaluate the ML model’s performance since they provide mean values for the sensitivity and specificity and recall and precision, respectively. Performance metrics are defined by Equations (1)–(6), where TP is a true positive (the correct prediction of a positive outcome), TN is a true negative (the correct prediction of a negative outcome), FP is a false positive (the wrong prediction of a negative outcome as positive), and FN is a false negative (the wrong prediction of a positive outcome as negative).
Precision = TP/(TP + FP)(1)
Recall = TP/(TP + FN)(2)
F1 Score = (2 × Precision × Recall)/(Precision + Recall)(3)
Balanced Accuracy (BA) = (Sensitivity + Specificity)/2(4)
where
Sensitivity = TP/(TP + FN)(5)

Specificity = TN/(TN + TP)(6)

The ML model training procedure was performed with each technique and each possible combination of stage 1. The implementation of feature fusion can be beneficial in enhancing the learning capabilities of the ML models, but non-important features can also lead to an increase in computational costs and the use of irrelevant information, which may affect the model’s performance. Therefore, limiting the number of features allows the ML models to focus on the most informative data by reducing model complexity and computational time [48,49]. To achieve this, RF-based feature selection is one of the most frequently employed techniques and demonstrates high performance compared to other approaches with a *FI* threshold of ≥0.05 [48,50]. Another feature selection approach is the calculation of *Shapley values* [51,52], which, as *FI*, measure each features’ impact on the output of the ML model [53]. However, the main difference between *FI* and *Shapley values* is that *FI* is computed during the training of the tree-based model (such as RF), whereas Shapley scores are based on Cooperative Game Theory and computed on fully trained models. Shapley values can assume both positive and negative values, and the sum of the values for a given instance (in this case, a single recording) may not necessarily total 100%. This is because the sum of all Shapley values for a given sample and a baseline value (expected output) represents an approximation of the degree of membership of that sample with regard to any of the prediction groups. Furthermore, to perform feature selection, the mean of all the Shapley values from each group across all instances is calculated and then a threshold is applied to decide which features are removed or retained.

### 2.4. Model Refinement

Considering this, after all the results were reported, the best performing model for each Flash Strength/Eye Concatenation was again passed through stage 3. This time, however, the selection of features was conducted in a comprehensive manner, employing two distinct approaches.

The first approach involved base feature selection on a *FI* threshold ≥ 0.01, where the *FI* was obtained by the model when it was tree-based (e.g., RF, AdaB, etc.) and if the model was kernel-based (MPL, SVM and KNN). The ≥0.01 threshold was selected due to the observation that using the standard 0.05 threshold resulted in the retention of only approximately 4 to 5 features, which significantly reduced the model’s ability to generalize.

The second approach involved Shapley values with a value threshold ≥ 0.005. A summary of the parameters and combinations utilized in the database is presented in Appendix A. It is noteworthy that, when employing multiple eyes or flash strengths, only the 113 and 446 Td.s flash strengths were utilized because the models fed with these two flash strengths consistently ranked within the top five. Accordingly, the decision was taken to only perform the concatenation with the ERG signal acquired with the two flash strengths that yielded the most promising results.

### 2.5. Statistics

Non-parametric tests (Wald, Chi-squared, or Kruskal–Wallis) were used as appropriate, with a *p*-value of <0.05 taken as significant. For pairwise comparisons, the *p*-value reported was subjected to post hoc analysis using the Dunn’s test with Holm–Bonferroni adjustment.

## 3. Results

### 3.1. Participants

The ERG dataset was collected at two sites—the Institute of Child Health at University College London (UCL) and the Flinders University in Adelaide, South Australia—across a period of five years from children and young adults with and without a neurodevelopmental condition. Participants from the Institute of Child Health were drawn from existing databases and those in Adelaide were recruited from the community. Participants were placed into ASD (*n* = 77), ADHD (*n* = 43), ASD + ADHD (*n* = 21), and control (*n* = 137) groups. The sex profile (male–female) of each group in terms of what was assigned at birth was as follows: ASD (56:17); ADHD (25:18); ASD + ADHD (16:5); control (57:80). There was a significant difference in sex between ASD and control (*p* < 0.001), ASD + ADHD, and control (*p* = 0.003) groups, but not between the ADHD and control groups (*p* = 0.058). We performed two-tailed Wald analysis. The profiles of each group in terms of the age in years (mean ± SD; range) were as follows: ASD (12.8 ± 4.3; 5.9–27.3); ADHD (13.0 ± 3.4; 6.2–21.8); ASD + ADHD (12.9 ± 4.4; 6.9–24.3); control (12.2 ± 4.5; 5.0–26.7). There were no significant group differences in the age of controls and either the ASD (*p* = 0.28), ADHD (*p* = 0.10) or ASD + ADHD (*p* = 0.46) groups or across all groups (*p* = 0.15), as determined using the Kruskal–Wallis test. See Appendix D, following the discussion section, for additional participant and site information.

The studies were approved by the South East Scotland Research Ethics Committee in the UK and by the Flinders University Human Research Ethics Committee and the Southern Adelaide Clinical Human Research Ethics Committee in Australia. Written informed consent was obtained from the parents/caregivers of children under the age of 16 or from the participants over the age of 16 who took part in the studies. Research complied with the tenets of the declaration of Helsinki.

### 3.2. Medication Effects

In both studies, participants were asked to refrain from taking any medications at least 24 h before testing, with the following medications used by the groups: melatonin (ASD *n* = 6, ADHD *n* = 3, ASD + ADHD *n* = 2, control *n* = 1); slow-release methylphenidate (ASD *n* = 2, ADHD *n* = 12, ASD + ADHD *n* = 4); D2-antagonist (ASD *n* = 3); risperidone-D2 and serotonin 5HT2A antagonists (ASD *n* = 2, ADHD *n* = 2, ASD + ADHD *n* = 1); Selective Serotonin Reuptake Inhibitors (ASD *n* = 5, ADHD *n* = 2, ASD + ADHD *n* = 1, control *n* = 1); Selective Norepinephrine Reuptake Inhibitors (ADHD *n* = 1); d-amphetamine (ADHD *n* = 6, ASD + ADHD *n* = 3); guanfacine (ADHD *n* = 1); and antihypertensives (ASD *n* = 2, ADHD *n* = 5, ASD + ADHD *n* = 2).

### 3.3. Between Groups Classification

Metrics for the ML models are reported in Table 1, Table 2, Table 3, Table 4, Table 5 and Table 6, with the best classification performance highlighted in bold for each of the comparisons.

Section 3.3.1, Section 3.3.2, Section 3.3.3 and Section 3.3.4 report the results obtained for the 2-group classification (control vs. ASD and control vs. ADHD), 3-group classification (control vs. ASD vs. ADHD), and the 4-group classification (control vs. ASD vs. ADHD vs. ASD + ADHD) both with and without medication and both including and excluding the influence of the TD features. For the 2-group classification, the participants that were in the ASD + ADHD group were included as part of the ASD or ADHD group.

#### 3.3.1. Two-Group Classification: ASD vs. Control

Table 1 presents the best model and the corresponding results for each flash strength/eye concatenation in the context of control vs. ASD classification. The XGB classifier, utilizing the concatenation of the TD + VFCDM features from the right and left eyes with the 446 Td.s flash strength, demonstrated the best performance for site 2 (UCL), with an F1 score of 0.761 and a BA of 0.759. Following feature selection, the XGB model exhibited a slight decline in performance, with a reduction in the F1 score to 0.748 and the BA to 0.747 when 35 rather than 136 features were used.

The impact of the features on the model outputs is summarized by the Shapley plot (Figure 2A for ASD and Figure 2B for ADHD). The Shapley plot shows the ten most impactful features regarding the output estimation. Each point shown in front of each feature represents a single instance from the dataset. The maximum value of each feature is shown as a red point, and the minimum value is shown as a blue point (also shown in the color bar on the right). In the Shapley summary, each point on the x-axis represents the Shapley score or contribution to the output for each sample for a particular class (in this case, the group indicated at the top of each figure). For example, in feature Tb_L446 in Figure 2A, higher values of the b-wave time to peak denoted (Tb), shown by some red and purple dots, can contribute positively to the membership score (in this case, the ASD group) with Shapley values between ~0.03 and 2.0, while smaller values of Tb contribute negatively, with Shapley values between ~0.05 and 2.1. The opposite is true for the feature vfcdm_kt_8_L446, where higher values of this feature contribute negatively to the membership score of the ASD group (and thus positively to the control group), while small values contribute positively (and thus negatively to the control group).

The Shapley values of the XGB indicated that Tb was the most important feature for the model, with a delay in Tb for the ASD group shown in the Shapley summary in Figure 2A. In addition, the VFCDM features at lower frequencies from the right eye and at higher frequencies from the left eye were of particular significance for ASD classification. When the VFCDM components were of a smaller magnitude, the likelihood that the individual was classified as belonging to the ASD group increased.

#### 3.3.2. Two-Group Classification: ADHD vs. Control

For ADHD classification compared to the control group at site 1 (Flinders), the RF classifier outperformed all the other models when using the TD + VFCDM + DWT feature concatenation of the ERG signal from the left and right eyes at 113 Td.s and 446 Td.s flash strengths. When the Shapley value feature selection was applied, reducing the features to 34 from 204, the F1 score and BA improved, with values of 0.773. See Table 2 for a summary of the performance of ADHD vs. control classification models and Figure 2B for the Shapley plot of the ADHD group classification. In this case, the VFCDM components at low frequencies (especially sub-band 3 and 4) demonstrated the highest importance for the RF classifier at the 133 Td.s flash strength. Other features that were also important included the high-frequency components (dwt_sum_OP160 at 446 Td.s and the dwt_bandhigh_10 at 113 Td.s).

#### 3.3.3. Three Group Classification: ASD vs. ADHD vs. Control

Table 3 reports the best results obtained for the three-group classification between ASD, ADHD, and controls. In this case, the KNN classifier achieved the highest performance when using the data from site 2 (UCL) and the combined TD + VFCDM + DWT features extracted from the ERG waveforms from both eyes at the strongest 446 Td.s flash strength. An F1 score of 0.660 and a BA of 0.704 were achieved with 41 out of 204 possible features when a feature selection criterion of ≥0.005 was applied to the Shapley values.

The Shapley summary of the ten most important features for the KNN model is shown in Figure 3 for the three-group comparison between control (3A), ASD (3B) and the ADHD (3C) groups. For the classification of these groups, the most important features were the kurtosis of the VFCDM 6th sub-band (ASD), the DWT OP160 value for control, and the DWT OP80 value for ADHD.

#### 3.3.4. Four Group Classification: ASD vs. ADHD vs. ASD + ADHD vs. Control

Table 4 shows the ML model’s performance when the ASD + ADHD group included participants meeting the DSM-5 diagnosis for both ASD and ADHD. Here, the RF classifier demonstrated the highest performance when data were acquired from the concatenation of all the 204 features of the ERG signals from the left eye using both the 113 and 446 Td.s flash strengths, with an F1 score of 0.477 and a BA of 0.491. However, when the number of features was reduced to 34 using *FI* selection, the F1 score increased to 0.526 and the BA increased to 0.529 using the left eye data at the single 446 Td.s flash strength.

Figure 4 shows the Shapley values for each of the four groups obtained with the RF classifier using the ERG signals from the left eye with the 446 Td.s flash strength. The ASD group maintained the same tendency, having low VFCDM and DWT feature values at higher frequencies (Figure 4A). In contrast to the 3-group classification, the features at higher frequencies became more important when classifying ADHD. This may be because of the co-occurrence of ASD + ADHD phenotypes in this group, with feature similarities shared between the ASD and ADHD groups. For example, shared features between the ASD + ADHD group and the ASD group included low values of the DWT OP80 summation (dwt_sum_OP80, *p* = 0.91); with the ADHD group, these included low values of the kurtosis of the 7th VFCDM component (vfcdm_kt_7, *p* = 0.21); with the control group, these included low values of the kurtosis of the 4th VFCDM component (vfcdm_kt_4, *p* = 0.37).

### 3.4. Effects of Medication

The influence of stimulant medications that elevate dopamine may affect the classification due to effects on the amplitude of the ERG [22,54]. Also see also Appendix A, which demonstrates the effect of methylphenidate on the amplitude of the ERG in an ADHD participant. For this reason, a comparison of the ML models’ performance was conducted, excluding the subjects who were taking medications from the dataset so that only participants that were medication-naïve were included. We excluded *n* = 6 ASD, *n* = 27 ADHD, and *n* = 13 ASD + ADHD participants from site 1 (Flinders) and *n* = 11 ASD, *n* = 16 ADHD, and *n* = 6 ASD + ADHD participants from site 2 (UCL).

### 3.5. Time Domain Feature Effects

The TD feature (Tb time to the peak of the b-wave) was of particular significance in the two-group and three-group classifications for ASD. However, when the medicated subjects were excluded, a single behavior was observed in the best model. The AdaB model exhibited overfitting to Tb and ignored the remaining features (see Appendix A). Although the F1 score was high when Tb was used, the mean AUC (0.65) was low. This became a problem for generalization, as the single value of Tb did not provide sufficient discriminatory information to distinguish ASD subjects from controls. Therefore, a secondary analysis was performed that *excluded* the TD features and only incorporated features from the signal analysis derived from VFCDM and DWT.

For the two-group classification (ASD vs. control) using the SVM model, the best performance was achieved at site 2 (UCL) with both flash strengths and eyes when the TD features were excluded but the medicated participants were included, with a BA and F1 score of 0.763 and a mean AUC of 0.83. For the two-group (ADHD vs. control) classification using the AdaB model, the best performance was achieved at site 1 (Flinders) when the TD features were included but the medicated participants were excluded, with an F1 score of 0.831, a BA of 0.842, and a mean AUC of 0.86. For the three-group classification (ASD vs. ADHD vs. control) using the KNN model, the best performance was achieved when including the TD and medicated participants from site 2 with both eyes and the 446 Td.s flash strength with an F1 score of 0.66, a BA of 0.704, and a mean AUC of 0.79. When the ASD + ADHD group was added, then the best performance was achieved using the RF classifier with the inclusion of all participants and the TD features from site 1 with the 446 Td.s flash strength, giving an F1 score of 0.526, a BA of 0.529, and a mean AUC of 0.73.

Table 5 summarizes the best ML model performance when the TD or medications were included in or excluded from the group classifications.

### 3.6. Effect of Sex

To explore the potential effects of sex assigned at birth on the ML performance metrics (with the exclusion of the TD features) for the ASD, ADHD, and control groups, the dataset was divided into male and female for each group. In the case of ASD classification, the best performance was obtained when only male participants were included with the AdaB model trained with participants from site 2 (UCL) with eye data from the 446 Td.s flash strength group. An F1 score of 0.873 with a mean AUC of 0.93 were obtained. In contrast, female-only classification was poorer, with an F1 score of 0.804 and a mean AUC of 0.77. In the case of ADHD classification, the RF model from site 2 with the 446 Td.s flash strength with both eyes performed best when classifying ADHD female-only participants with an F1 score of 0.840 and a mean AUC of 0.89. However, the performance metrics were not significantly different to the classification with male-only assessments or when both sexes included. See Table 6 for a summary of the results.

There were differences in the importance of the flash strength for each sex and group, with the 446 Td.s being the most important for ASD classification and the 113 Td.s being the most important for ADHD classification for male participants. In contrast, for female participants, the pattern was reversed (See Figure 5). For female participants, all the top 10 features were statistically significant (*p* < 0.05) based on the Kruskal–Wallis test, but this was not true for the male participants, as highlighted in the Shapley summary plot (Figure 5). There were some differences based on sex for the group classification. Regarding male sex, for the ASD group (Figure 5A) and the ADHD group (Figure 5B), the higher-frequency components (VFCDM sub-bands 7 and 8) were most important. In contrast, for female ASD (Figure 5C) and ADHD (Figure 5D) groups, the lower-frequency components (VFCDM sub-bands 2 and 3) were the most important for classification.

### 3.7. Individual Case Analysis

The ML models support a group-level classification, with caveats related to sex, medication use, and the inclusion of the TD features that modify the model’s performance. It is possible to explore the feature importance that leads to the individual classification on a case-by-case basis. This approach may be useful for stratifying phenotypes based on sub-classes, as has been performed with fMRI datasets [55,56]. To demonstrate this potential next step, we present waterfall plots that display the individual explanation of the features for an individual case (ASD vs. control) and (ADHD vs. control) with the explained expected model output (*E[f(X)]*) according to the information from the entire feature dataset. Here, each feature drives the classification in either a negative or a positive direction towards the ASD or ADHD classification. When the final explained output *f(x)* is greater than *E[f(X)]*, then the individual is classified as belonging to either ASD or ADHD in this example.

Individual-level ASD classification is shown in Figure 6, where it is compared with the results of an individual from the control group. The most important features in this case identifying the ASD individual was the kurtosis of the 7th VFCDM sub-band (240–280 Hz) with the 446 Td.s flash strength (vfcdm_kt_7_R446) (*p* < 0.05). Here, according to the summary plot, the lower the feature value, the greater the positive contribution to the output. However, in some cases, participants had similar features, even if they were not from the same group. In the example of the ASD individual, this feature had a value of 2.638, giving a contribution of +0.11, while for the control participant, the value was 4.562, giving a positive impact, but with a smaller contribution of +0.03. Finally, the explained outputs for the control and ASD participants were *f(x)* = 0.281 and *f(x)* = 0.857 with *E[f(X)]* = 0.472.

In the case of ADHD classification, the AdaB classifier was more sensitive to boundary values, namely, fixed contribution values that were assigned to the features, as shown in the waterfall plot in Figure 7. For example, the interquartile value of the 2nd VFCDM sub-band with the 446 Td.s flash strength applied to the right eye (vfcdm_iqr_2_R446) (*p* < 0.05) was the feature with the most significant impact on the output, especially when it was present at large values. However, the ADHD and control participants obtained similar values for this feature, so that in both cases, even for ADHD, the contribution moved towards the control group phenotype, decreasing the total positive contribution of the 105 features. This illustrates that many of the features did not have a major impact on the output, as only eight of the total features were used by the AdaB classifier after feature selection. Finally, the explained outputs for the control and ADHD participants were *f(x)* = 0.448 and *f(x)* = 0.593, with *E[f(X)]* = 0.555.

## 4. Discussion

The findings support the potential use of ML to identify individuals that meet diagnostic criteria for ASD, ADHD, or ASD + ADHD from a typically developing control population. The ERG, as a direct measure of central nervous system activity that can be recorded in childhood non-invasively, can help to identify individuals that have a different neurodevelopmental trajectory. However, different ML strategies were required to classify each group depending on their sex, medication use, flash strength, and the inclusion or exclusion of TD features in the classification task. To summarize, the models for ASD or control classification performed best when the TD features were excluded and only male participants were included, resulting in an F1 score of 0.873, a BA of 0.87 and a mean AUC of 0.93 (Table 7). Conversely, for female participants, the F1 score and BA were lower (0.804 and 0.814), with a mean AUC of 0.77. In the case of ASD, excluding medicated participants did not improve the performance metrics of the ML models with an F1 score of 0.744 and a mean AUC of 0.73 when TD features and participants using medications were excluded. Thus, the main factor affecting ASD classification was sex, with better results for male participants and after the influence of the TD features was removed.

In contrast, for the ADHD or control classification, the best performance was achieved when participants taking medications were excluded but the TD features were included, with an F1 score of 0.831, a BA of 0.842, and a mean AUC of 0.86 (See Table 6). With respect to sex, the best classification occurred with female ADHD participants without the TD features, with an F1 score, a BA of 0.840, and a mean AUC of 0.89 (See Table 7). With respect to the three-group classification between ASD, ADHD, and control, an equivalent performance was achieved with or without the medicated participants and with or without the TD features with an F1 score of 0.660, a BA of 0.704, and a mean AUC of 0.79. For the final case, which included those with a co-occurring diagnosis of ASD + ADHD, the best performance achieved was an F1 score of 0.526, a BA of 0.529, and a mean AUC of 0.73. This analysis included the TD features and medicated participants. The significant drop in performance in the 4-group classification compared to the 3-group classification, with an F1 score of 0.526, was most likely due to the small number of ASD + ADHD participants and the phenotypic overlap with the participants with a sole ASD or ADHD diagnosis. This highlights the complexities of differentiating between heterogeneous neurodevelopmental conditions [57,58]. Thus, the ML approach can support classification between ASD or ADHD and a typically developing control population but is less robust when including all three groups or participants with an ASD and ADHD diagnosis.

Following a systematic evaluation of ML models for group classification, involving the fusion of different ERG waveform features and exploring the combinations with sex, site, eye, and flash strength, and the inclusion and exclusion of medicated subjects and the TD features, we obtained results. The results of all the tests demonstrated that, regarding this task, no specific model exhibited a consistently higher performance across all k-group evaluations. However, the incorporation of feature fusion and feature selection proved to be a crucial element in the performance of most of the models. In terms of improving the ML modeling, the implementation of feature fusion plus feature selection improved the F1 score in some cases, such as for ADHD vs. control. The F1 score increased from 0.786 to 0.831 when the number of features included in the ML model was reduced from 136 to 8. In addition, the use of Shapley values as a feature selection approach outperformed the tree-based *FI* approach for the two- and three-group classifications (see Table 5, where the BA is greater when Shapley values are used instead of *FI*). A possible reason for this was that the *FI* approach, even with a lower threshold than commonly used, still excluded more features than necessary, discarding features that may have been informative for some participants during the classification.

However, the Shapley values enabled the visualization of the most important features used for classification, which varied for each group. For the two-group classification, the time to peak of the b-wave was most important feature for ASD classification, while for ADHD classification, the kurtosis of the third VFCDM sub-band (80–120 Hz) was the most important feature. However, improved classification was obtained for the ASD group when the TD features were excluded and, in addition, the model’s performance was best when only applied to male participants. This may be due to the higher proportion of male ASD individuals and the higher likelihood of being diagnosed with ASD if male [59]. Here, the best feature to classify ASD males was the kurtosis of the 280–320 Hz sub-band from the VFCDM analysis in the 446 Td.s flash strength. The high flash strength feature supports the involvement of the ON pathway as a marker for ASD, as previously observed, with reduced rod-driven ON pathway responses also low in ASD populations [10,12]. For ADHD classification, excluding participants that were using medications and using features from TD and VFCDM yielded the best results with features from a combination of the 113 and 446 Td.s strengths and each eye. However, in the case of ADHD classification, the result was slightly better for females than for either males or both sexes combined. This may indicate that the ERG is more sensitive for female ADHD participants and supports previous work that found that female ADHD individuals had more reduced a- and b-wave amplitudes than male individuals [20]. Table 7 summarizes the best performances for each of the models evaluated.

### 4.1. Medications

The exploration of the inclusion/exclusion of the medicated subject revealed that the use of medications can influence the strength of the classification models for ADHD. Given the use of methylphenidate to manage symptoms in ADHD [60] patients and its potential confounding effects on retinal signaling [22,61], future studies will need to ensure an adequate period of drug washout. The results showed that excluding medicated subjects improved the overall classification performance in ADHD. One potential future direction would be to investigate if the ERG signal can be used to determine the efficacy of any medications in ADHD that target dopamine, such as methylphenidate, in a comparable manner to the Qbtest [62]. Appendix A shows the differences in the TD parameters between sites with the non-medicated participants, where there is large inter-site variability in the b-wave amplitude and time to peak (Tb). Further controlled studies will be needed to elucidate the contribution of medications to the variability in the ERG TD parameters.

### 4.2. Inter-Site Variability

The combined datasets from site 1 and site 2 did not result in improved classification for either the ASD or ADHD group. This was most likely due to the differences in the diagnostic procedures used for recruitment at each site. Site 1 (Flinders) relied upon documented community-based assessments that were based on an observational and parental interview, with support in some cases by questionnaires, to form the diagnosis. In contrast, participants at site 2 (UCL) were seen in a tertiary setting and had a more formal diagnostic process with the use of ADOS and the 3Di to support the final diagnosis [63,64]. This discrepancy between sites was most notable in the amplitude of the b-wave within the ADHD group. The UCL cohort had high b-waves and this was potentially the result of a more stringent diagnostic process given that the Flinders ADHD group tended to have low b-wave amplitudes (see Appendix A). It should be noted that the low b-wave amplitudes were also reported in ADHD cases, where the formal diagnosis was based on community services [20]. This added heterogeneity in the ADHD populations from each site led to a negative impact on the recognition of the neurodevelopmental groups, especially the ADHD and ASD + ADHD groups, where most of the subjects were misclassified as either ASD or control, as shown in Appendix A. An additional test was conducted by removing the ADHD group from Flinders, which was the group most affected by these discrepancies, and combining the remaining subjects from the two sites. However, no significant changes in the performance were evidenced, with F1 scores of approximately 0.73, 0.565, and 0.45, for the 2-group (ADHD), 3-group, and 4-group classifications, respectively.

### 4.3. Sex

The division and testing of the data based on sex highlighted the insight that sex affects the ERG waveform and consequently the extracted features. According to the results and the Shapley values ranking, the most important and statistically significant information from female subjects was concentrated in bands of certain frequencies. For example, for ASD females, the model was mostly focused on high-frequency bands, whereas for ADHD females, the features that had more impact to the output were in the low-frequency bands. The role of sex in neurodevelopmental conditions is an area of interest, with genetic and environmental factors likely to play a role [65,66,67].

## 5. Conclusions

This study is a first step towards the development of identifying an ERG-based biomarker for diagnosing neurodevelopmental disorders. However, several limitations still remain, including the inter-site variability in our study. This highlights the need for standardized diagnostic assessments across different clinical and research settings with larger, multi-site studies needed to fully assess the generalizability of these findings. Additionally, longitudinal studies are needed to understand how any ERG characteristics may change over the course of development or with long-term medication use. The integration of ERG data with other potential biomarkers, such as attention and impulsivity markers as assessed with the Qbtest [68], pupil responses [69], blink rate [70], or eye-tracking [71], could also enhance diagnostic accuracy and provide a more comprehensive understanding of neurodevelopmental differences across the neurodevelopmental spectrum of ASD and ADHD.

Recording an ERG in children younger than five years of age is possible but represents challenge, especially in children that may be non-verbal, a factor which may limit the application of the current protocol. However, the development of innovative smartphone-based devices [72,73,74] may enable the ERG to be recorded more readily in younger population when combined with signal analysis and ML modeling, potentially allowing us to screen for individuals with a different neurodevelopmental profile.

The signal analysis of the ERG offers a way to explore multiple features contained within the TD signal to support classification models in neurological [5,16,24,25,26,45,75,76] and retinal disorders [77,78,79,80]. Other approaches to the remodeling of the ERG waveform may also prove beneficial. An example of this involves applying the principles of Functional Data Analysis (FDA) [81,82] to reveal differences in the waveform shape based on a registered time series [83]. The FDA approach could provide a more nuanced analysis of the ERG waveform shapes, potentially capturing subtle differences that are not clear in traditional methods of time series analysis for disease classification [84]. In addition, the augmentation of the natural dataset with AI-generated synthetic waveforms may also improve dataset balancing and size to enhance classification models [85]. Deep learning models using neural networks may also offer advances in this field, offering larger datasets [86]. These advanced analytical approaches, combined with larger and more diverse participant cohorts, have the potential to significantly advance our understanding of retinal function in neurodevelopmental disorders and move us closer to clinically useful diagnostic biomarkers [1].

## Figures and Tables

**Figure 1 bioengineering-12-00015-f001:**
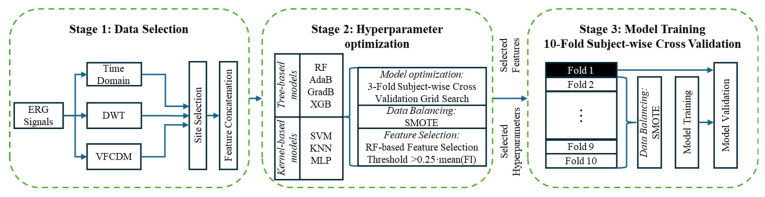
Model training procedure flowchart incorporating three stages. One: the data selection of the electroretinogram signal with time–frequency analysis (discrete wavelet transform (DWT) and variable-frequency complex demodulation (VFCDM)) and standard time domain features. Two: hyperparameter optimization, including machine learning model optimization, data balancing with synthetic minority oversampling (SMOTE), and optimal feature selection before. Three: model training with a 10-fold subject-wise cross validation. Models: RF (Random Forest); AdaB (Adaptive Boosting); GradB (Gradient Boosting); XGB (Extreme Gradient Boosting); SVM (Support Vector Machine); KNN (K-nearest neighbor); MLP (multi-layer perceptron).

**Figure 2 bioengineering-12-00015-f002:**
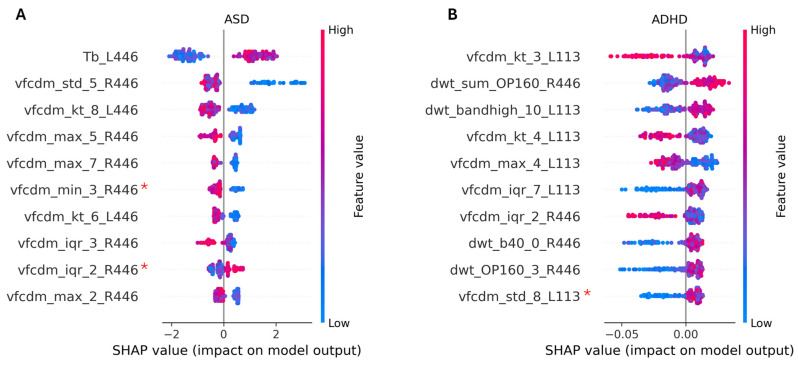
Shapley summary of the ten most important features for the XGB classifier for ASD (**A**) and the RF classifier for ADHD (**B**). For ASD, the time of the b-wave peak at 446 Td.s (left eye) (Tb_L446) was the most important feature in the model in terms of differentiating the ASD group from the control group. For the ADHD group, the kurtosis of the 3rd band at 113 Td.s (left eye) (vfcdm_kt_3_L113) had the highest Shapley feature importance for classification. A red asterisk (*) refers to statistically non-significant differences (*p* ≥ 0.05), as assessed based on the Kruskal–Wallis test, between the groups for these features.

**Figure 3 bioengineering-12-00015-f003:**
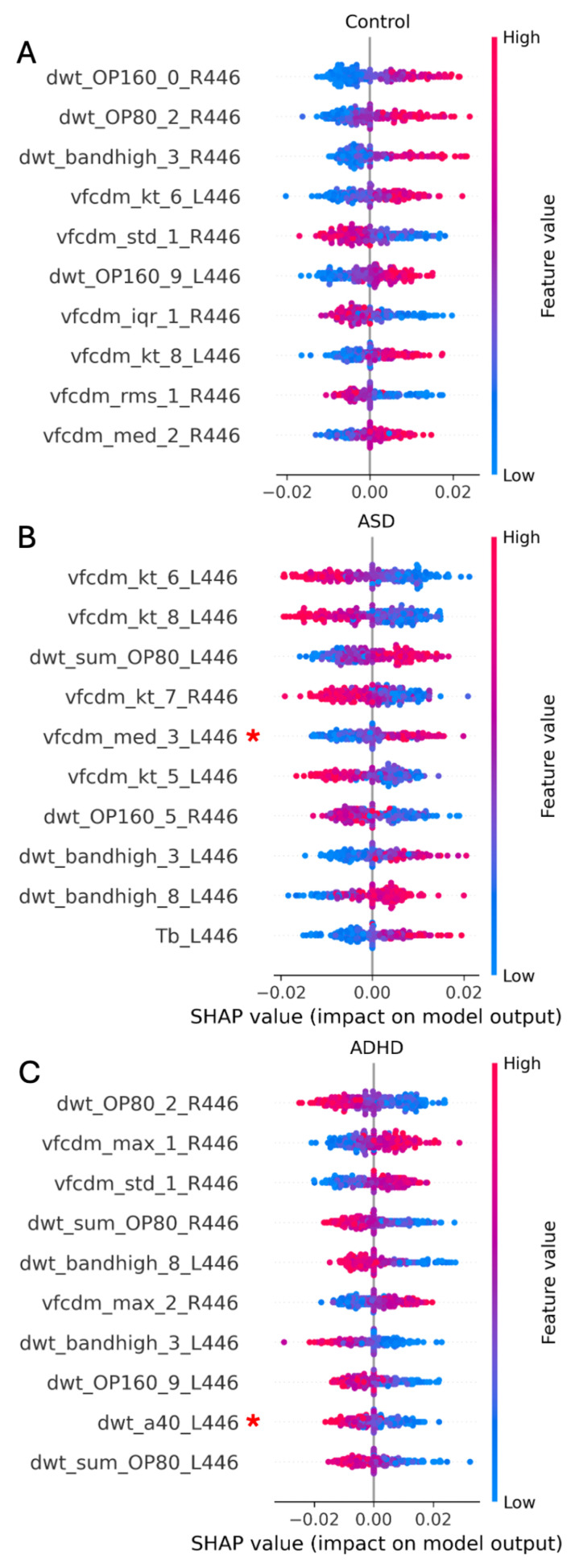
A Shapley summary of the top 10 features for KNN classifier for the 3-group classification: (**A**) control, (**B**) ASD, and (**C**) ADHD. Feature importance for the group classification is indicated by the red scale. The most important features were as follows: for control, it was the DWT OP160 component; for ASD, it was the kurtosis of the 6th VFCDM sub-band; and for ADHD, it was the DWT OP80 component. The red asterisk (*) indicates a statistically non-significant difference (*p* ≥ 0.05) between the groups for these features based on the Kruskal–Wallis test.

**Figure 4 bioengineering-12-00015-f004:**
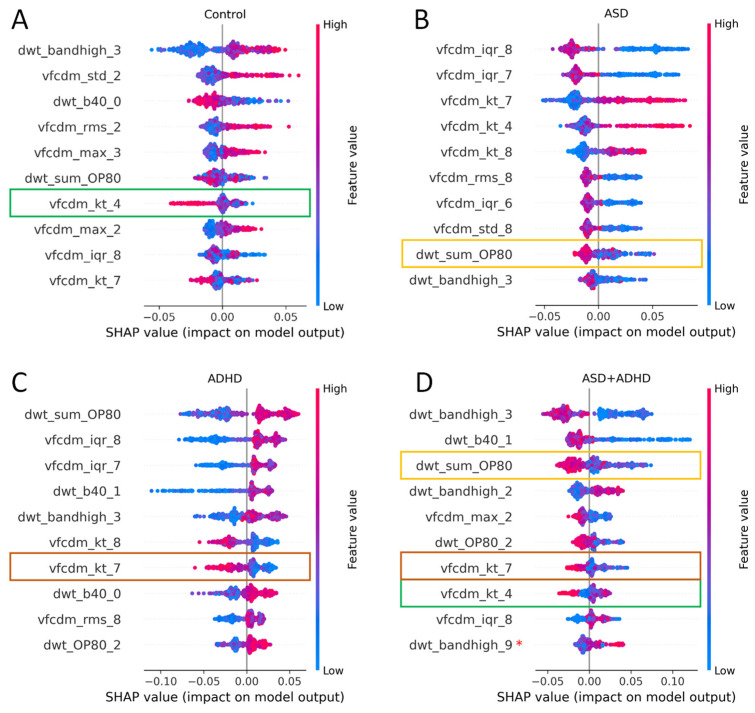
A Shapley summary of the top 10 features of the RF classifier for the 4-group classification: control (**A**), ASD (**B**), ADHD (**C**), and ASD + ADHD (**D**). When classifying between the four groups, the most important features were as follows: for the control, it was the high-frequency DWT range; for ASD, it was the interquartile range of the 8th VFCDM sub-band; for ADHD, it was the sum of the OP80 DWT components; for ASD + ADHD, it was also the high-frequency DWT range. Red markers in the Shapley plots have greater importance for classifying the group. The colored boxes highlight the common important features for each of the group’s classifications highlighted vfcdm_kt_4 (green), vfcdm_kt_7 (red) and dwt_sum_OP80 (yellow). The red asterisk (*) refers to statistically non-significant (*p* ≥ 0.05) differences between the groups for these features based on the Kruskal–Wallis test.

**Figure 5 bioengineering-12-00015-f005:**
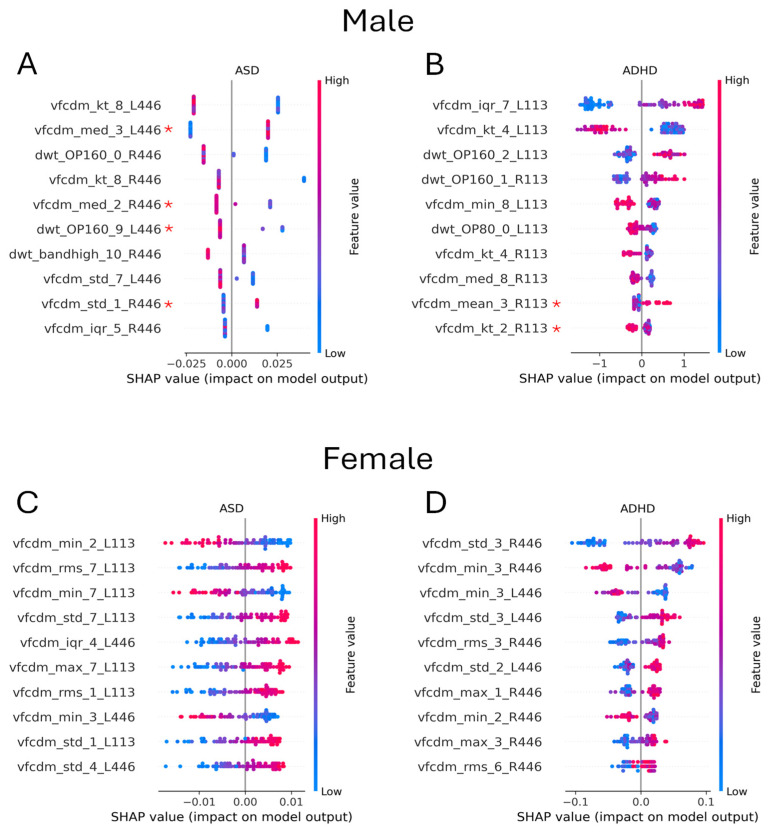
A Shapley summary of the best models for male or female 2-group classification (ASD vs. control or ADHD vs. control). (**A**,**B**) show the male participants for ASD or ADHD vs. control, respectively, and (**C**,**D**) show the female participants for ASD or ADHD vs. control, respectively. For males, the high-frequency features from the VFCDM were more important for ASD and ADHD classification (sub-bands 7 and 8). In contrast, for female participants, the lower-frequency features were more important for ASD or ADHD classification (sub-bands 2 and 3). A red asterisk (*) refers to statistically non-significant differences (*p* ≥ 0.05) based on the Kruskal–Wallis test between the groups for these features.

**Figure 6 bioengineering-12-00015-f006:**
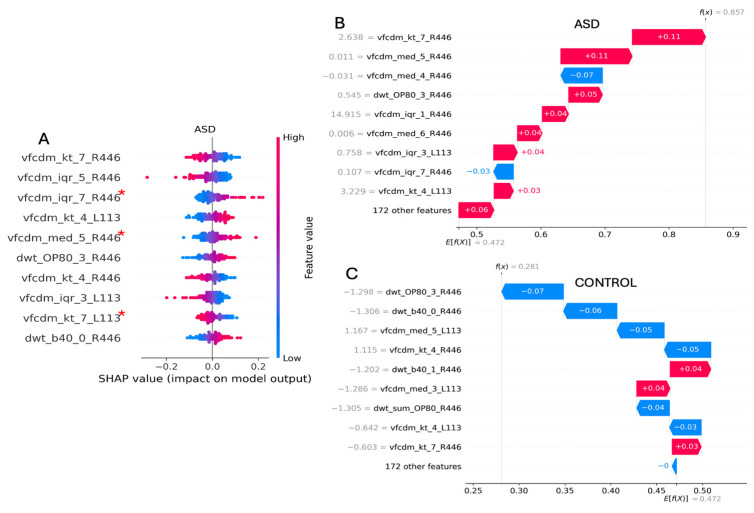
(**A**) Shapley values of the SVM classifier that achieved the best 2-group (ASD vs. control) classification performance. Waterfall plots for an individual control (**C**) and an individual ASD (**B**) participant with the relative contributions to a positive classification for control (blue) and ASD (red). At the left side of each feature name, the actual value of the feature from that subject is presented. A red asterisk (*) refers to statistically non-significant differences (*p* ≥ 0.05) based on the Kruskal–Wallis test between the groups for these features.

**Figure 7 bioengineering-12-00015-f007:**
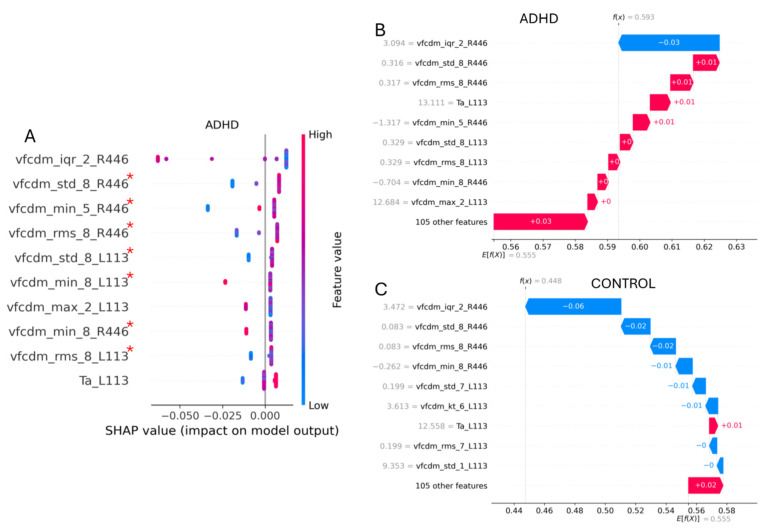
(**A**) Shapley values of the SVM classifier that achieved the best 2-group (ADHD vs. control) classification performance. Waterfall plots for an individual control (**C**) and an individual ADHD (**B**) participant with the relative contributions to a positive classification for control (blue) and ADHD (red). At the left side of each feature name, the actual value of the feature from that subject is presented. A red asterisk (*) refers to statistically non-significant differences (*p* ≥ 0.05) based on the Kruskal–Wallis test between the groups for these features.

**Table 1 bioengineering-12-00015-t001:** The best model performance for each type of flash strength/eye concatenation assessment for ASD classification compared to the control group.

Technique	Model	Site	Eye Flash Strength	# Samples [Control/ASD]	BA	F1 Score	# Features
TD + VFCDM + DWT	AdaB	2	R-446	[83/73]	0.712	0.712	102
TD + VFCDM	XGB	2	R-446/L-446	[80/62]	**0.759**	**0.761**	**136**
Selected Features	KNN	2	L-113/L-446	[77/60]	0.727	0.726	36
TD + VFCDM (*FI* ≥ 0.01) *	XGB	2	R-446/L-446	[77/60]	0.747	0.748	35
TD + VFCDM (*Shapley val* ≥ 0.005) *	XGB	2	R-446/L-446	[77/60]	0.745	0.747	96

*BA* = *Balanced Accuracy*; FI = feature importance; R = right eye; L = left eye; strength = flash strength (Td.s); val = value; # = number. * the top 1 model architecture of each concatenation was trained using exhaustive feature selection using feature importance and Shapley values. Only the best performing models are shown.

**Table 2 bioengineering-12-00015-t002:** Best performances in each type of flash strength/eye concatenation for ADHD classification compared to the control.

Technique	Model	Site	Eye-Strength	# Samples [Control/ADHD]	BA	F1 Score	# Features
TD + DWT	XGB	1	L-113	[122/74]	0.750	0.750	38
TD + VFCDM + DWT	SVM	1	R-446/L-446	[116/69]	0.724	0.726	204
TD + VFCDM + DWT	RF	1	L-113/R-446	[112/67]	0.758	0.760	204
TD + VFCDM + DWT (*FI* ≥ 0.01) *	RF	1	L-113/R-446	[112/67]	0.727	0.729	23
TD + VFCDM + DWT (*Shapley val* ≥ 0.005) *	RF	1	L-113/R-446	[112/67]	**0.773**	**0.773**	**34**

*BA* = *balanced accuracy*; *FI* = feature importance; R = right eye; L = left eye; strength = flash strength (Td.s); val = value; # = number. * the top 1 model architecture of each concatenation was trained using exhaustive feature selection via feature importance and Shapley values, with only the best performance shown for the model architecture. We used a combination of time domain (TD), discrete wavelet transform (DWT), and variable-frequency complex demodulation (VFCDM) features.

**Table 3 bioengineering-12-00015-t003:** Best performances in each type of flash strength/eye concatenation for 3-group classification between the ASD, ADHD, and control groups.

Technique	Model	Site	Eye Strength	# Samples [Control/ASD/ADHD]	BA	F1 Score	# Features
TD + VFCDM + DWT	GradB	1	L-446	[128/50/55]	0.581	0.578	102
TD + VFCDM + DWT	KNN	2	R-446/L-446	[80/47/24]	0.672	0.622	204
Selected Features	SVM	1	L-113/L-446	[115/47/51]	0.648	0.620	36
TD + VFCDM + DWT (*FI* ≥ 0.01) *	KNN	2	R-446/L-446	[80/47/24]	0.610	0.579	18
TD + VFCDM + DWT (*Shapley val* ≥ 0.005) *	KNN	2	R-446/L-446	[80/47/24]	**0.704**	**0.660**	**41**

*BA* = *balanced accuracy*; *FI* = feature importance; R = right eye; L = left eye; strength = flash strength (Td.s); val = value; # = number. * the top 1 model architecture of each concatenation was trained using exhaustive feature selection using feature importance and Shapley values. Only the best performance is shown in those cases. For the three-group classification the combined features VFCDM, DWT, and time domain (TD) analyses provided the best classification using both eyes and the second site’ data.

**Table 4 bioengineering-12-00015-t004:** Best performances in each type of flash strength/eye concatenation for 4-group classification between control, ASD, ADHD, and ASD + ADHD.

Technique	Model	Site	Eye Strength	# Samples [Control/ASD/ADHD/ASD + ADHD]	BA	F1 Score	# Features
TD + VFCDM + DWT	RF	1	L-446	[128/50/55/21]	0.468	0.474	102
TD + VFCDM	KNN	2	R-446/L-446	[80/47/24/15]	0.477	0.461	136
TD + VFCDM + DWT	RF	1	L-113/L-446	[115/47/51/20]	0.491	0.477	204
TD + VFCDM + DWT (*FI* ≥ 0.01) *	RF	1	L-446	[128/50/55/21]	**0.529**	**0.526**	**34**
TD + VFCDM + DWT (*Shapley val* ≥ 0.005) *	RF	1	L-446	[128/50/55/21]	0.521	0.517	31

*BA* = *balanced accuracy*; *FI* = feature importance; R = right eye; L = left eye; strength = flash strength (Td.s); val = value; # = number. * the top 1 model architecture of each concatenation was trained using exhaustive feature selection using feature importance (*FI*) or Shapley values. Only the best performance is shown from each model of the data.

**Table 5 bioengineering-12-00015-t005:** A comparison of the best models including and excluding time domain features and medicated subjects.

Technique	Model	Site	Eye Strength	# Samples	# Feats	BA	F1 Score	Mean AUC	# Groups	TD	Med
TD + VFCDM	XGB	2	R-446/L-446	[80/62]	136	0.759	0.761	0.78	2 (ASD)	Y	Y
VFCDM + DWT (*Shapley val* ≥ 0.005)	SVM	2	L-113/R-446	[77/60]	123	**0.763**	**0.763**	**0.83**	2 (ASD)	**N**	**Y**
TD + DWT	AdaB	2	R-113/L-446	[75/46]	1	0.730	0.729	0.65	2 (ASD)	Y	N
VFCDM + DWT (*Shapley val* ≥ 0.005)	SVM	2	R-446/L-446	[78/47]	54	0.738	0.744	0.73	2 (ASD)	N	N
TD + VFCDM + DWT (*Shapley val* ≥ 0.005)	RF	1	L-113/R-446	[112/67]	34	0.773	0.773	0.81	2 (ADHD)	Y	Y
VFCDM (*Shapley val* ≥ 0.005)	SVM	2	R-113/R-446	[79/44]	62	0.809	0.801	0.86	2 (ADHD)	N	Y
TD + VFCDM (*Shapley val* ≥ 0.005)	AdaB	1	L-113/R-446	[112/31]	8	**0.842**	**0.831**	**0.86**	2 (ADHD)	**Y**	**N**
VFCDM(*Shapley val* ≥ 0.005)	AdaB	2	R-446/L-446	[78/19]	9	0.817	0.799	0.84	2 (ADHD)	N	N
TD + VFCDM + DWT (*Shapley val* ≥ 0.005)	KNN	2	L-446/R-446	[80/47/24]	41	**0.704**	**0.660**	**0.79**	3	**Y**	**Y**
VFCDM + DWT (*Shapley val* ≥ 0.005)	SVM	1	R-113/L-446	[110/45/52]	47	0.649	0.641	0.81	3	N	Y
TD + VFCDM (*FI* ≥ 0.01)	GradB	1	L-113/R-446	[112/41/23]	18	0.662	0.635	0.80	3	Y	N
VFCDM + DWT (*FI* ≥ 0.01)	XGB	1	R-446/L-446	[116/42/23]	30	0.604	0.609	0.78	3	N	N
TD + VFCDM + DWT (*FI* ≥ 0.01)	RF	1	L-446	[128/50/55/21]	34	**0.529**	**0.526**	**0.73**	4	**Y**	**Y**
VFCDM + DWT (*Shapley val* ≥ 0.005)	RF	1	L-446	[128/50/55/21]	45	0.510	0.499	0.72	4	N	Y

BA = balanced accuracy; *FI* = feature importance; Med = medication; N = no; R = right eye; L = left eye; Strength = flash strength; TD = time domain, (Td.s); val = value; Y = yes; # = number. Group 2: control vs. ASD or control vs. ADHD. Group 3: control vs. ASD vs. ADHD. Group 4: control vs. ASD vs. ADHD vs. ASD + ADHD. The TD features included the a-wave and b-wave time and amplitude. The medications classification included those using medications and those not using medications.

**Table 6 bioengineering-12-00015-t006:** Results by sex. All the models were trained without using TD features.

Technique	Model	Site	Eye Strength	# Samples	# Feats	BA	F1 Score	Mean AUC	# Groups	Sex
VFCDM + DWT (*Shapley val* ≥ 0.005)	SVM	2	R-113/L-446	[77/60]	123	0.763	0.763	0.82	ASD vs. Con	Both
VFCDM + DWT (*FI* ≥ 0.01)	AdaB	2	R-446/L-446	[39/50]	26	**0.870**	**0.873**	**0.93**	ASD vs. Con	Male
VFCDM (*Shapley val* ≥ 0.005)	SVM	2	R-113/L-446	[40/11]	10	0.814	0.804	0.77	ASD vs. Con	Female
VFCDM *(Shapley val* ≥ 0.005)	SVM	2	R-113/L-446	[79/44]	62	0.809	0.801	0.86	ADHD vs. Con	Both
VFCDM + DWT (*FI* ≥ 0.01)	XGB	1	R-113/L-113	[37/47]	32	0.793	0.794	0.86	ADHD vs. Con	Male
VFCDM	RF	2	R-446/L-446	[41/18]	128	**0.840**	**0.840**	**0.89**	ADHD vs. Con	Female

BA = balanced accuracy; *FI* = feature importance; R = right eye; L = left eye; strength = flash strength (Td.s); val = value; # = number. # Samples: ASD = autism spectrum disorder; Con = control; ADHD = attention deficit/hyperactivity disorder.

**Table 7 bioengineering-12-00015-t007:** The summary table of the best performances in each test.

Technique	Model	Site	Eye Strength	# Samples	# Feats	BA	F1 Score	Mean AUC	Sex	TD	Med
**2-Group ASD vs. Control Classification**											
VFCDM + DWT (*Shapley val* ≥ 0.005)	SVM	2	L-113/R-446	[77/60]	123	0.763	0.763	0.83	Both	N	Y
VFCDM + DWT (*FI* ≥ 0.01)	AdaB	2	R-446/L-446	[39/50]	26	0.870	0.873	0.93	Male	N	Y
**2-Group ADHD vs. Control Classification**											
TD + VFCDM (*Shapley val* ≥ 0.005)	AdaB	1	L-113/R-446	[112/31]	8	0.842	0.831	0.86	Both	Y	N
VFCDM	RF	2	R-446/L-446	[41/18]	128	0.840	0.840	0.89	Female	N	Y
**3-Group ASD vs. ADHD vs. Control Classification**											
TD + VFCDM + DWT (*Shapley val* ≥ 0.005)	KNN	2	L-446/R-446	[80/47/24]	41	0.704	0.660	0.79	Both	Y	Y
**4-Group ASD vs. ADHD vs. ASD + ADHD vs. Control Classification**											
TD + VFCDM + DWT (*FI* ≥ 0.01)	RF	1	L-446	[128/50/55/21]	34	0.529	0.526	0.73	Both	Y	Y

BA = balanced accuracy; TD = time domain; Med = medication.

## Data Availability

The datasets presented in this article are not readily available because the data are part of ongoing analyses. Requests to access the datasets should be directed to the corresponding author.

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
