# Peer review of "Spectral Analysis of Light-Adapted Electroretinograms in Neurodevelopmental Disorders: Classification with Machine Learning"

_bioengineering, 2024, doi:10.3390/bioengineering12010015_

Round 1
Reviewer 1 Report
Comments and Suggestions for Authors
I thank the authors for the opportunity to review the article “Spectral Analysis of the Light-Adapted Electroretinogram in Neurodevelopmental Disorders: Classification with Machine Learning”. The article employs seven ML methods to identify three populations with neurodevelopmental disorders: ASD, ADHD, and ASD+ADHD from a control population. The findings suggest that different ML methods do well in classifying ASD and ADHD, but not a combination of classes. However, the article is lacking in several areas. These areas should be addressed before publication:
1. The manuscript does not include a separate literature review section. Thus, the authors should expand the introduction to discuss key studies in the field. This should include an analysis of prior findings and how they relate to the current study. Without this, readers cannot fully appreciate the background against which the research is situated.
2. The manuscript does not discuss the methods commonly used in the field of ML on healthcare/neurodevelopmental disorders and their efficacy in addressing the research question. Add a detailed discussion of the current methods to address the problem, including their strengths and weaknesses. Perhaps a summary table of existing works, their features, ML method used and accuracy/metrics.
3. The authors do not clearly articulate the gap in existing research that necessitates this work. Multiple works in the current literature cover ML/DL/AI methods for neurodevelopmental disorders. What is introduced in this study that separates this work from others?
4. The Materials and Methods section is poorly structured, making it challenging to follow the sequence of actions or activities performed during the study. The sequence of steps taken to perform the study is unclear, leaving readers confused about how the research was conducted from start to finish with the possibility of replicating. A chronological presentation of the process is necessary to ensure coherence.
5. Some subsections in sections 2 and 3 are convoluted and inundates the readers with much information. The authors can consider presenting these in tables so that it is easy on the eye and also clarity of presentation.
6. The presentation of figures need improvement in two areas: (1) size needs to be larger, (2) resolution needs improvement, (3) the figure caption needs to describe the figures clearly, especially when there are multiple subfigures. Proper referring of these subfigures is appreciated.
This reviewer has no issues with the methodology employed and the results presented. However, the weakness of a poor introduction and methodology section requires significant edits. In particular, the authors need to address the gap in existing research. Without the context of previous work conducted in this field/domain, it is difficult to evaluate the impact of this work’s results vis-à-vis other state-of-the-art technologies and methods.
Author Response
- The manuscript does not include a separate literature review section. Thus, the authors should expand the introduction to discuss key studies in the field. This should include an analysis of prior findings and how they relate to the current study. Without this, readers cannot fully appreciate the background against which the research is situated.
A section on the background and previous works is now provided in the Introduction:
“The origins of harnessing the retinal signals as a potential biomarker in ASD date back to 1988 when Edward Ritvo first explored the dark adapted ERG responses in n=27 children and adults with ASD and found a reduced b-wave amplitude in ap-proximately half the participants which suggested a glutamate signaling pathways deficit in this group [10]. Ritvo also performed a small pilot study in a family but did not identify a strong familial link with the ERG responses [11]. It was much later that these initial findings were extended by Constable et al., (2016) [12] in a small adult population of high functioning adults with ASD which supported these initial findings and further identified reductions in the light-adapted ERG b-wave amplitudes and shape of the oscillatory potentials (OPs).
The first multicenter study to be performed in (n=90) ASD and (n=87) control children and revealed a reduced a- and b-wave amplitude at the higher flash strength, however the study did not explore the OPs in detail [13] but found the PhNR was also normal suggesting normal retinal ganglion cell function [14]. One observation during this study was that some children who presented had a co-occurring diagnosis of ASD+ADHD and it was noted that this sub-group had elevated b-waves compared to the children with a sole ASD diagnosis. This led to an exploratory study in (n=15) ADHD participants that revealed a large b-wave amplitude in this group that differentiated ADHD from ASD [15]. Signal analysis of the available waveforms supported higher energy in the ERG signal in the ADHD participants from this study [16]. These findings supported the conclusions of Lee et al., (2022) [15] that the pattern of differ-ences in the ERG signal were most likely related to difference in the balance of GABA and glutamate signaling in ASD and ADHD. However, Friedel and colleagues failed to replicate the findings of a reduced b-wave in (n=32) ASD adults using the same ERG recording protocol [17]. Huang et al., (2024) [18] also reported no significant difference in the b-wave amplitude in ASD adults, but a larger a-wave which was reduced following a GABA(B) agonist suggesting the ERG may provide a possible pathway to monitor drug efficacy in ASD [18].
With respect to ADHD, there is evidence for greater background retinal ‘neural noise’ in (n=20) adults that correlates with inattention scores [19] supporting a functional change in the retina’s of ADHD individuals. Most recently ERG findings in (n=27) ADHD adults did not replicate the elevated b-waves previously reported but instead highlighted a reduced light adapted a- and b-wave amplitude and delayed b-wave time to peak amongst the female ADHD participants which suggested some sex difference within the ADHD population [20]. Taken together the different studies have reported contrary results at times which may be due to several factors such as the diagnostic procedures at different locations, sex, or age of the populations. Most stud-ies until more recently have relied upon the time-domain features of the ERG such as the amplitude and time to peak of the principal components or the shape and peak of the photopic hill and its mathematical characterization [21]. The effects of methylphenidate, used to control ADHD symptoms and elevates central dopamine levels also shows changes in the ERG amplitude in some participants [22].”
- The manuscript does not discuss the methods commonly used in the field of ML on healthcare/neurodevelopmental disorders and their efficacy in addressing the research question. Add a detailed discussion of the current methods to address the problem, including their strengths and weaknesses. Perhaps a summary table of existing works, their features, ML method used and accuracy/metrics.
A comprehensive review of the numerous studies is perhaps beyond the scope of this paper but we have added a paragraph discussing imaging, EEG and eye movement studies with the main limitation being that prior studies have looked at two classes ASD vs Control and in this case we apply ML models to three classes of neurodevelopmental conditions. Added references are also noted:
“The use of ML models has led to a large number of studies that have explored potential biomarkers in neurodevelopmental disorders- with a comprehensive review beyond the scope of this study. Review of imaging studies report accuracies of between 48.3-97% for ASD classification with heterogeneity in the study population [39, 40]. Attempts have been made to address this issue using the Autism Brain Imaging Data Exchange dataset acquired from several sites with an area under the curve of 79% obtained based on brain network analysis [41]. Eye movement studies based on gaze patterns in ASD with ensemble ML algorithms report an F1-score of 95.5% and provides a fast and user-friendly method for ASD classification [42]. Eye tracking has been successfully used in children under three years of age with Random Forest classifier to obtain an area under the curve of 0.85 using Shapley feature selection [43]. Electroencephalogram recordings have also yielded classification accuracies of 98% when neural networks were employed to classify ASD [44]. Some limitations of these studies is that the models support classification of two groups such as ASD vs Control and do not consider other neurodevelopmental disorders or individuals that meet an ASD and ADHD diagnosis and so the classification accuracies are limited to one class.”
Added references:
Xu M, Calhoun V, Jiang R, Yan W, Sui J. Brain imaging-based machine learning in autism spectrum disorder: methods and applications. J Neurosci Methods. 2021 Sep 1;361:109271. doi: 10.1016/j.jneumeth.2021.109271. Epub 2021 Jun 24. PMID: 34174282; PMCID: PMC9006225.
Liu, M., Li, B., & Hu, D. (2021). Autism Spectrum Disorder Studies Using fMRI Data and Machine Learning: A Review. Frontiers in neuroscience, 15, 697870. https://doi.org/10.3389/fnins.2021.697870
Ingalhalikar, M., Shinde, S., Karmarkar, A., Rajan, A., Rangaprakash, D., & Deshpande, G. (2021). Functional Connectivity-Based Prediction of Autism on Site Harmonized ABIDE Dataset. IEEE transactions on bio-medical engineering, 68(12), 3628–3637. https://doi.org/10.1109/TBME.2021.3080259
Sá, R. O. D. S., Michelassi, G. C., Butrico, D. D. S., Franco, F. O., Sumiya, F. M., Portolese, J., Brentani, H., Nunes, F. L. S., & Machado-Lima, A. (2024). Enhancing ensemble classifiers utilizing gaze tracking data for autism spectrum disorder diagnosis. Computers in biology and medicine, 182, 109184. https://doi.org/10.1016/j.compbiomed.2024.109184
Wei, Q., Dong, W., Yu, D., Wang, K., Yang, T., Xiao, Y., Long, D., Xiong, H., Chen, J., Xu, X., & Li, T. (2024). Early identification of autism spectrum disorder based on machine learning with eye-tracking data. Journal of affective disorders, 358, 326–334. https://doi.org/10.1016/j.jad.2024.04.049
Ranaut, A., Khandnor, P., & Chand, T. (2024). Identifying autism using EEG: unleashing the power of feature selection and machine learning. Biomedical physics & engineering express, 10(3), 10.1088/2057-1976/ad31fb. https://doi.org/10.1088/2057-1976/ad31fb
- The authors do not clearly articulate the gap in existing research that necessitates this work. Multiple works in the current literature cover ML/DL/AI methods for neurodevelopmental disorders. What is introduced in this study that separates this work from others?
Whilst it is true that there are studies using ML Deep learning models to classify ASD or ADHD – this study has data across these two diagnostic categories and included participants with ASD + ADHD as a diagnosis and so attempts to classify these three diagnostic groups based on ERG signal analysis using raw data and manipulations of the ML models through different feature selection approaches.
The following sentence has been added to the concluding paragraph of the introduction.
“This study includes diagnostic groups including ASD, ADHD and individuals that meet both ASD and ADHD diagnostic criteria to elucidate if ML models with feature refinements in feature selection can classify across the three categories from a typically developing population and as such provides insights into the strengths and weaknesses of ML models for this task.”
4.The Materials and Methods section is poorly structured, making it challenging to follow the sequence of actions or activities performed during the study. The sequence of steps taken to perform the study is unclear, leaving readers confused about how the research was conducted from start to finish with the possibility of replicating. A chronological presentation of the process is necessary to ensure coherence.
We have reviewed this section and attempted to clarify the steps further – and acknowledge that there is a lot of information – we have moved some tables in line with the AE’s suggestions and modified the headings of each section to provide a little more clarity about each section. Our aim was to provide as much information regarding the study populations and study protocols for the reader. We have confined the materials and methods section to include a description of the ML methods and ERG recordings with participant information moved to the results as suggested by reviewer 3 – which may make this section more focussed.
5. Some subsections in sections 2 and 3 are convoluted and inundates the readers with much information. The authors can consider presenting these in tables so that it is easy on the eye and also clarity of presentation.
We have added additional information to each Table caption and provide a summary Table of the key findings in the Discussion to aid the reader. The tables are presented to provide all of the key findings for the reader.
6.The presentation of figures need improvement in two areas: (1) size needs to be larger, (2) resolution needs improvement, (3) the figure caption needs to describe the figures clearly, especially when there are multiple subfigures. Proper referring of these subfigures is appreciated.
Figures have been enlarged in the MS with a reworking of figures 3 , 6 and 7. Captions have also been amended to provide more detail and each figure part is referenced in the main manuscript.
This reviewer has no issues with the methodology employed and the results presented. However, the weakness of a poor introduction and methodology section requires significant edits. In particular, the authors need to address the gap in existing research. Without the context of previous work conducted in this field/domain, it is difficult to evaluate the impact of this work’s results vis-à-vis other state-of-the-art technologies and methods.
Thank you we have tried to amend as much as possible with these kind suggestions for improvements.
Reviewer 2 Report
Comments and Suggestions for Authors
Authors present a Spectral Analysis of the Light-Adapted Electroretinogram in Neurodevelopmental Disorders and provide a Classification with Machine Learning. In the focus are patients with autism spectrum disorder (ASD) and ADHD. Technical part of the manuscript needs evaluation by a reviewer with experience in bioengineering. Potential clinical application of these findings as well as future advances of physiological biomarkers and its applications for clinic are needed.
Author Response
Authors present a Spectral Analysis of the Light-Adapted Electroretinogram in Neurodevelopmental Disorders and provide a Classification with Machine Learning. In the focus are patients with autism spectrum disorder (ASD) and ADHD. Technical part of the manuscript needs evaluation by a reviewer with experience in bioengineering. Potential clinical application of these findings as well as future advances of physiological biomarkers and its applications for clinic are needed.
We thank the reviewer for their thoughts, and we feel that the technical parts of the manuscript have been reviewed by reviewers 1 and 3.
We have attempted to address the future potential of the ERG and now include a small section on feasibility based on reviewer 3’s suggestions (Point 22). We include a list of other new biomarkers in this field including eye tracking and pupil responses and now also include blink rate as a new potential biomarker that may prove more effective and discuss future directions with synthetic waveform generation and deep learning model to enhance classification in the conclusions. discuss the current use of ML in retina: Reference added:
Krishnappa Babu, P. R., Aikat, V., Di Martino, J. M., Chang, Z., Perochon, S., Espinosa, S., Aiello, R., L H Carpenter, K., Compton, S., Davis, N., Eichner, B., Flowers, J., Franz, L., Dawson, G., & Sapiro, G. (2023). Blink rate and facial orientation reveal distinctive patterns of attentional engagement in autistic toddlers: a digital phenotyping approach. Scientific reports, 13(1), 7158. https://doi.org/10.1038/s41598-023-34293-7
Reviewer 3 Report
Comments and Suggestions for Authors
The current manuscript uses machine learning approaches to see whether different models can be trained using electroretinograms (ERGs) to classify individuals with either autism spectrum disorder (ASD, n=77), attention deficit hyperactivity disorder (ADHD, n=43) or comorbid ASD+ADHD (n=21) from control subjects (n=137).
I have some comments that would strengthen the article.
Minor comments:
1, Avoid starting a sentence with an abbreviation i.e., line 63, 67, 607 etc.
Keywords
2, Line 39 (and line 78): Suggest using gender instead of sex.
Introduction
3, Line 51: Authors should clarify what Appendix A refers too. It was not part of the supplementary section.
4, Line 53-60: I suggest the authors make this section more digestible to the reader. It comes across quite technical. Could the authors provide a bit more explanation how machine learning (ML) is used to generate classification models using ERG waveform signal analysis in the time-frequency domain? I understand this is somewhat covered in the methods section but some mention of it would be useful.
5, Line 69-71: Can the authors expand on their aims a bit more? In line 70 what are the classifications in neurodevelopmental disorders (i.e. ASD vs. ADHD, ASD+ADHD vs. controls) is the text referring too? Why is this important in the context of the current study?
Methods
Section 2.1:
6, The participant demographics (gender, age) should go into the results section and not be in the methods. A two tailed Wald is also mentioned without any explanation of the statistical tests used in the manuscript.
7, Line 75: Please provide more description about the two sites. I am aware this is mentioned in section 4.2, but some description of the sites should also be added before.
8, Line 85: Could not find Appendix B (appendices are not listed in the supplementary material either).
Section 2.2:
9, As above, section 2.2 forms parts of the results and should be placed as part of the results section.
10, The medications should be clarified “D2-antagonist (ASD=3)”, “serotonin 5HT2A antagonist” (line 96), SSRI (line 97) and SNRI (line 98).
Section 2.3:
10, I suggest the authors clarify what is meant by Td.s (elsewhere written as TD [line 225]). Are ‘Td.s’ and ‘TD features’ the same?
Section 2.4:
11, Line 125: Please clarify the location of Appendix C
Section 2.5
12, Line 129: Explain why ‘seven’ ML learning techniques were conducted. This seems a bit arbitrary to me.
13, Line 134: The seven ML techniques should be incorporated into Figure 1 and the figure ought to be modified to show how these 7 ML techniques were divided into the three stages.
14, Line 151: A sentence explaining what ‘hyperparameter optimisation’ is would improve the readability of the manuscript for readers not familiar with this term.
15, Line 163: The authors should clarify why a 10-fold subject-wise cross-validation approach was used.
16, Line 167-168: More information i.e. a rationale on why the F1-score and balanced accuracy (BA) metrics were selected to evaluate the ML models performance. A clearer explanation of what is meant if the performance metrics were ‘true negative’ or ‘true positive’ should also be stated.
17, Line 187: I suggest the authors provide some more explanation what is meant by a ‘feature’ in the context paper.
Results
18, Table 1: As per my comment in #16 above what is the threshold for a clinically meaningful result for the BA and F1 score? I note that the BA, F1 score and # features values in Table 4 are bolded for the technique of TD+VFCDM using the XGB model, however, there is no explanation why these values are in bold. This also applies to other tables i.e. Table 2, 3 etc.
19, Line 250: I suggest the authors clarify what is meant by Tb in the context of features. This appears rather abruptly without having been mentioned previously.
20, Children with ASD can also present with autonomic dysregulation and one of the symptoms is pupillary dilation. Did the authors consider this in the context of the study?
Discussion
21, In the beginning of the discussion, could the authors have a couple of sentences to explain the key findings of the manuscript without using ML terminology/ I am aware that ML is a blossoming area of research, but all too often it gets complicated with terms and different notions. Having a couple of sentences to explain the clinical utility of the findings I feel would be of benefit to the wider readership of the journal.
Conclusions
22, The conclusion could be shortened and should focus how feasible would such an approach be in real-world settings for diagnosing children with ASD from ADHD etc. For example, (I) do the authors consider the approach would be feasible in children with severe ASD? (II) What age would be best? (III) What would the potential limitations of such an approach be in individuals with multiple comorbid conditions?
Ethics
23, One of the sites was in London, England – can the authors clarify why Southeast Scotland Research Ethics Committee was used?
Author Response
Minor comments:
1, Avoid starting a sentence with an abbreviation i.e., line 63, 67, 607 etc.
Sentences now begin with a non-abbreviated word such as:
Line 63 Applications using ML….
Line 67: Several ML models….
Line 607 A FDA approach….
Line 726 The ADHD diagnoses were….
Line 717 The ASD participants
Line 821 The DWT analysis….
Keywords
2, Line 39 (and line 78): Suggest using gender instead of sex.
We appreciate the comment and have defined now in the manuscript that ‘sex’ refers to the biological sex assigned at birth as we did not specifically record the gender preference term for this cohort and for the purposes of the ‘report’ we confine ourselves to the use of sex in relation to the sex assigned at birth and not the social construct of ‘gender’.
Line 71 the influence of sex assigned at birth
Line 78 The sex profile (male:female) assigned at birth
Introduction
3, Line 51: Authors should clarify what Appendix A refers too. It was not part of the supplementary section.
The following note is added to the reference of each Appendix. Note that with the restyructure of the text Appendix B has been moved to become Appendix C (with Appendix C now Appendix B).
Line 51 see Appendix A (following the discussion section)…
Line 167 See Appendix B following the discussion section for additional participant and site information
Line 304 in Appendix C (following the discussion section
4, Line 53-60: I suggest the authors make this section more digestible to the reader. It comes across quite technical. Could the authors provide a bit more explanation how machine learning (ML) is used to generate classification models using ERG waveform signal analysis in the time-frequency domain? I understand this is somewhat covered in the methods section but some mention of it would be useful.
The introduction to this paragraph has been modified to now include an explanation of the added value of signal analysis in the context of developing ML models that we hope will address this point from line 93.
“The ERG is conventionally analyzed using the main time domain parameters that relate to the amplitude of the two main a- and b-wave peaks and the time points at which the peaks occur [23]. However, relying on these markers limits the number of features that ML models could use to improve classification between groups. Therefore, signal analysis in the time-frequency domain has been utilized to decompose the signal into multiple frequency based sub-bands [16, 24-26] which expands the number of features available for ML models. The time-frequency analysis of the ERG waveform was developed by Gauvin and colleagues who decomposed the ERG signal using a discrete wavelet transform (DWT) with a Haar mother wavelet [27-30]. An alternative signal analytical approach using variable frequency complex demodulation (VFCDM) [31] has been applied to the ERG waveforms and compared to DWT with the introduction of ML for group classification with the models achieving a sensitivity of 0.85 and specificity of 0.78 [24, 26].”
5, Line 69-71: Can the authors expand on their aims a bit more? In line 70 what are the classifications in neurodevelopmental disorders (i.e. ASD vs. ADHD, ASD+ADHD vs. controls) is the text referring too? Why is this important in the context of the current study?
We have added some more background to the genetic and common occurrence of ASD and ADHD to support the exploration of the three groups to the introduction:
“Several studies have explored ML models to improve biomarker discovery in neurodevelopmental disorders- with a comprehensive review beyond the scope of this study. Review of imaging studies report accuracies of between 48.3-97% for ASD classification with heterogeneity in the study population [39, 40]. Attempts have been made to address this issue using the Autism Brain Imaging Data Exchange dataset ac-quired from several sites with an area under the curve of 79% obtained based on brain network analysis [41]. Eye movement studies based on gaze patterns in ASD with ensemble ML algorithms report an F1-score of 95.5% and provides a fast and us-er-friendly method for ASD classification [42]. Eye tracking has been successfully used in children under three years of age with Random Forest classifier to obtain an area under the curve of 0.85 using Shapley feature selection [43]. Electroencephalogram recordings have also yielded classification accuracies of 98% when neural networks were employed to classify ASD [44]. Some limitations of these studies is that the models support classification of two groups such as ASD vs Control and do not consider other neurodevelopmental disorders or individuals that meet an ASD and ADHD diagnosis and so the classification accuracies are limited to one class.”
Additional References
Martinez, S., Stoyanov, K., & Carcache, L. (2024). Unraveling the spectrum: overlap, distinctions, and nuances of ADHD and ASD in children. Frontiers in psychiatry, 15, 1387179. https://doi.org/10.3389/fpsyt.2024.1387179
Berg, L. M., Gurr, C., Leyhausen, J., Seelemeyer, H., Bletsch, A., Schaefer, T., Pretzsch, C. M., Oakley, B., Loth, E., Floris, D. L., Buitelaar, J. K., Beckmann, C. F., Banaschewski, T., Charman, T., Jones, E. J. H., Tillmann, J., Chatham, C. H., Bourgeron, T., EU-AIMS LEAP Group, Murphy, D. G., … Ecker, C. (2023). The neuroanatomical substrates of autism and ADHD and their link to putative genomic underpinnings. Molecular autism, 14(1), 36. https://doi.org/10.1186/s13229-023-00568-z
Methods
Section 2.1:
6, The participant demographics (gender, age) should go into the results section and not be in the methods
This section along with medications is now at the beginning of the results section as requested
A two tailed Wald is also mentioned without any explanation of the statistical tests used in the manuscript.
A note on statistics is now included in the Methods section with the tests Wald described in the text when used.
“2.5. Statistics
Non-parametric tests (Wald, Chi-squared or Kruskal-Wallis) were used as ap-propriate with a p-value of <0.05 taken as significant. For pairwise comparison’s the p-value reported followed post hoc analysis using the Dunn’s test with Holm-Bonferroni adjustment.“
7, Line 75: Please provide more description about the two sites. I am aware this is mentioned in section 4.2, but some description of the sites should also be added before.
The following introduction has been provided to this section:
“The ERG dataset was collected at two sites: the Institute of Child Health at University College London (UCL) and the second at Flinders University in Adelaide, South Australia across a period of five years in children and young adults with and without a neurodevelopmental condition. Participants from the Institute of Child Health (were drawn from existing databases and those in Adelaide were recruited from the community.”
8, Line 85: Could not find Appendix B (appendices are not listed in the supplementary material either).
Appendix’s follow the Discussion and are now ‘sign posted’ – We had regionally included the appendices as part of the supplementary material, but the policy was to have all citations in the reference list and so the material is appended with references to the main manuscript as Appendices. The supplementary material contains just additional figures and descriptions.
Section 2.2:
9, As above, section 2.2 forms parts of the results and should be placed as part of the results section.
Medication section is now moved to the Results section along with the Participant information.
10, The medications should be clarified “D2-antagonist (ASD=3)”, “serotonin 5HT2A antagonist” (line 96), SSRI (line 97) and SNRI (line 98).
Here we grouped the trade names by their main action rather than listing the various brand names and dosages as an overview. We have expanded on the SSRI and SNRI and indicated that the methylphenidate was in slow-release form to give a little more information.
Section 2.3:
10, I suggest the authors clarify what is meant by Td.s (elsewhere written as TD [line 225]). Are ‘Td.s’ and ‘TD features’ the same?
TD is Time domain and is defined on lien 137:
…was the fusion of the Time Domain (TD) features
Td.s is a Troland second and a unit of retinal illuminance per unit time. The following definition is added at line 126.
Flash strength is reported in Troland.seconds (Td.s).
Section 2.4:
11, Line 125: Please clarify the location of Appendix C
The following line has been added:
“Appendix C (following the discussion section..”
Section 2.5
12, Line 129: Explain why ‘seven’ ML learning techniques were conducted. This seems a bit arbitrary to me.
Our main idea was to perform a comprehensive evaluation of different machine learning models, since, depending on the task, some of them perform better than others, due to the variety of nonlinear and complex patterns that they can learn. The following addition has been made to the introduction of the Machine Learning section in the Methods:
“A systematic evaluation of seven ML techniques was conducted that included: Random Forest (RF), Adaptive Boosting (AdaB), Gradient Boosting (GradB), Extreme Gradient Boosting (XGB), Support Vector Machine (SVM), K-nearest neighbor (KNN) and multi-layer perceptron (MLP). This was done to consider the great variety of non-linear and complex patterns that each technique can learn and the variation in performance that this may imply. The training of each model was divided into three stages (dataset selection, hyperparameter optimization and model training) with each stage outlined in Figure 1.”
13, Line 134: The seven ML techniques should be incorporated into Figure 1 and the figure ought to be modified to show how these 7 ML techniques were divided into the three stages.
Figure 1 is intended as a simplified overview of the three stages for the ML techniques. So, we added all the models within the figure, as all the models follow the same training procedure, as mentioned in line~ 274.
“The ML model training procedure was performed with each technique and each possible combination of stage 1. “
The caption to Figure 1 has been expanded to give a more detailed description:
“Figure 1. Model training procedure flowchart incorporating three stages including One: data se-lection of the electroretinogram signal with time-frequency analysis (Discrete Wavelet Transform (DWT) and Variable Frequency Complex Demodulation (VFCDM)) and standard Time Domain features. Two: Hyperparameter optimization including machine learning model optimization, data balancing with Synthetic Minority Oversampling (SMOTE) and optimal feature selection before and finally three: model training with a 10-fold subject-wise cross validation. Models included: RF Random Forest, AdaB Adaptive Boosting, GradB Gradient Boosting, XGB Extreme Gradient Boosting, SVM Support Vector Machine, KNN K-Nearest Neighbor and MLP Multi-Layer Perceptron.”
14, Line 151: A sentence explaining what ‘hyperparameter optimisation’ is would improve the readability of the manuscript for readers not familiar with this term.
The following definition or explanation is inserted to provide further information on the process:
“Hyperparameter optimization is the process of identifying the optimal hyperparameter values, which define the learning of ML models. This is typically accomplished by training and testing the model on multiple subsets, varying the values, and selecting the ones that yield the best performance.”
15, Line 163: The authors should clarify why a 10-fold subject-wise cross-validation approach was used.
In line ~216, a brief explanation is given:
“In clinical prediction models, cross-validation should be done on a subject basis for a meaningful evaluation, as in our study where multiple samples per subject exist. This helps to improve the generalization and results interpretation of the model [47]. In addition, this prevents data leakage, where a subject's data is used both for training and testing. Furthermore, due to the limited number of instances for certain groups (such as ASD+ADHD) and to ensure sufficient variation within the training set, a k=10 was selected for the k-fold cross-validation. At each fold, the training set was subjected to SMOTE balancing to maintain the same number of instances per group during the training. However, the test set kept the unbalanced distribution. As a result of the im-balance problem, the F1-score and balanced accuracy (BA) were selected as the most appropriate evaluation metrics to evaluate the ML models performance, since they provide a mean value for the sensitivity and specificity and of recall and precision respectively.
With the following reference added:
Saeb, S., Lonini, L., Jayaraman, A., Mohr, D. C., & Kording, K. P. (2017). The need to approximate the use-case in clinical machine learning. Gigascience, 6(5), gix019. https://doi.org/10.1093/gigascience/gix019
16, Line 167-168: More information i.e. a rationale on why the F1-score and balanced accuracy (BA) metrics were selected to evaluate the ML models performance. A clearer explanation of what is meant if the performance metrics were ‘true negative’ or ‘true positive’ should also be stated.
The explanation of the use of F1-Score and Balanced accuracy, along with the True negative and positive samples is given in line ~221
“However, the test set kept the unbalanced distribution. As a result of the imbalance problem, the F1-score and balanced accuracy (BA) were selected as the most appropri-ate evaluation metrics to evaluate the ML models performance, since they provide a mean value for the sensitivity and specificity and of recall and precision respectively. Performance metrics are defined by Equations 1-6 where TP is True Positive (correct prediction of a positive outcome), TN is True Negative (correct prediction of a negative outcome), FP is False Positive (wrong prediction of a negative outcome as positive), and FN is False Negative (wrong prediction of a positive outcome as negative).”
17, Line 187: I suggest the authors provide some more explanation what is meant by a ‘feature’ in the context paper.
The following definition or explanation is inserted at line 168 to ‘explain’ what is meant by ‘a feature’
“A Feature in this context is defined as measurement obtained from the ERG signal through TD or time-frequency domain analysis.”
Results
18, Table 1: As per my comment in #16 above what is the threshold for a clinically meaningful result for the BA and F1 score? I note that the BA, F1 score and # features values in Table 4 are bolded for the technique of TD+VFCDM using the XGB model, however, there is no explanation why these values are in bold. This also applies to other tables i.e. Table 2, 3 etc.
A clarification of the results highlighted in bold is included in the introduction to the Results
“Metrics for the ML models are reported in Tables 1-6 with the best classification performance highlighted in bold for each of the comparisons.”
19, Line 250: I suggest the authors clarify what is meant by Tb in the context of features. This appears rather abruptly without having been mentioned previously.
Tb is now redefined when it occurs in the results section as we acknowledge that this term is not common and can be re-emphasized.
“Figure 2A, higher values of the b-wave time to peak denoted (Tb) (red dots and some purple)…”
20, Children with ASD can also present with autonomic dysregulation and one of the symptoms is pupillary dilation. Did the authors consider this in the context of the study?
Pupil diameter does not influence the retinal response – because the recording method measures pupil diameter before and during recording so that the flash stimulus is corrected/adjusted for pupil diameter – hence the Troland unit which depends upon pupil diameter. In the future we may consider investigating the pupillary light response but as a factoir in the analysis pupil diameter was not included as the ERG response was independent of pupil diameter.
Discussion
21, In the beginning of the discussion, could the authors have a couple of sentences to explain the key findings of the manuscript without using ML terminology/ I am aware that ML is a blossoming area of research, but all too often it gets complicated with terms and different notions. Having a couple of sentences to explain the clinical utility of the findings I feel would be of benefit to the wider readership of the journal.
The following sentence has been added as on the one hand ML models can help with classification but with some limitations but on a higher level the ERG could simply identify children that have a different CNS profile that may indicate ‘some’ DSM formal diagnosis of ASD/ADHD. We note that DSM classification of these neurodevelopmental conditions have shifted and changed over the years and maybe it is ‘all ‘ we can do at this stage is simply identify children with an altered retinal signal that ‘might’ indicate some neurological difference.
“The ERG as a direct measure of Central Nervous System activity that can be recorded in childhood non-invasively could help in identifying individuals that have a different neurodevelopmental trajectory.”
Conclusions
22, The conclusion could be shortened and should focus how feasible would such an approach be in real-world settings for diagnosing children with ASD from ADHD etc. For example, (I) do the authors consider the approach would be feasible in children with severe ASD? (II) What age would be best? (III) What would the potential limitations of such an approach be in individuals with multiple comorbid conditions?
The following section has been added to the Conclusions to address feasibility-
“The feasibility of recording an ERG in children younger than five years of age is possible though a challenge – and especially in children that may be non-verbal which may limit the current protocol. However, developments of innovative smartphone-based devices [72-74] may enable the ERG to be recorded more readily in younger population when combined with signal analysis and ML modelling to potentially screen for or identify individuals with a different neurodevelopmental profile.”
Added References:
- Tuesta, R., Harris, R., Posada-Quintero H.F. Circuit and sensor design for smartphone-based electroretinography. IEEE 20th International Conference on Body Sensor Networks (BSN) 2024, 2024, 1-4. doi:10.1109/BSN63547.2024.10780451
- Cordoba, N., Daza, S., Constable, P.A., Posada-Quintero H.F. Design of a smartphone-based clinical electroretinogram recording system. IEEE International Symposium on Medical Measurements and Applications (MeMeA) 2024, 1-2. doi:10.1109/MeMeA60663.2024
- Huddy, O., Tomas, A., Manjur, S.M., Posada-Quintero H.F. Prototype for Smartphone-based Electroretinogram. IEEE 19th International Conference on Body Sensor Networks (BSN). 2023, 1-4. doi:10.1109/BSN58485.2023.10330910
Ethics
23, One of the sites was in London, England – can the authors clarify why Southeast Scotland Research Ethics Committee was used?
Yes, this seems to be a quirk of the Ethics committee that related to the South East of England and Scotland. We have updated the ethics committees to provide greater detail – but ‘yes’ this is the name of the ethics committee.
Ethics section now reads:
“Institutional Review Board Statement: The studies were conducted in accordance with the Decla-ration of Helsinki. Protocol 2 was approved in Australia by the Southern Adelaide Clinical Human Research Ethics Committee (Approval Code: 318.16) and the Flinders University Human Research and Ethics Committee (Approval Code: 4606). Protocol 1 was approved by the Flinders University Human Research and Ethics Committee (Approval Code: 7180) and the South east Scotland Research Ethics Committee in the UK (Approval Code:18/SS/0008).”
Round 2
Reviewer 1 Report
Comments and Suggestions for Authors
The authors have satisfactorily addressed my comments. The manuscript is now of the quality worthy of publication.
Author Response
We thank the reviewer
Reviewer 2 Report
Comments and Suggestions for Authors
Sufficient response to my remarks.
Author Response
We thank teh reviewer
Reviewer 3 Report
Comments and Suggestions for Authors
Thank you for improving the manuscript.
Author Response
Thank you for your comments and support